# The Fire Inventory from NCAR version 2.5: an updated global fire emissions model for climate and chemistry applications

Christine Wiedinmyer[1], Yosuke Kimura[2], Elena McDonald-Buller[2], Louisa K. Emmons[3], Rebecca R. Buchholz[3], Wenfu Tang[3], Keenan Seto[1], Maxwell B. Joseph[1,4], Kelley C. Barsanti[5,3], Annmarie G. Carlton[6], and Robert Yokelson[7]

[1]Cooperative Institute for Research in Environmental Sciences, University of Colorado Boulder, Boulder, CO, USA
[2]Center for Energy and Environmental Resources, The University of Texas at Austin, Austin, TX, USA
[3]Atmospheric Chemistry Observations and Modeling Laboratory, National Center for Atmospheric Research, Boulder, CO, USA
[4]Earth Lab, University of Colorado Boulder, Boulder, CO, USA
[5]Department of Chemical & Environmental Engineering, Center for Environmental Research & Technology, University of California Riverside, Riverside, CA, USA
[6]Department of Chemistry, University of California Irvine, Irvine, CA, USA
[7]Department of Chemistry, University of Montana, Missoula, MT, USA

*Correspondence to*: Christine Wiedinmyer (Christine.Wiedinmyer@colorado.edu)

**Abstract.** We present the Fire Inventory from NCAR version 2.5 (FINNv2.5), a fire emissions inventory that provides publicly available emissions of trace gases and aerosols for various applications, including use in global and regional atmospheric chemistry modeling. FINNv2.5 includes numerous updates to the FINN version 1 framework to better represent burned area, vegetation burned, and chemicals emitted. Major changes include the use of active fire detections from the Visible Infrared Imaging Radiometer Suite (VIIRS) at 375 m spatial resolution, which allows smaller fires to be included in the emissions processing. The calculation of burned area has been updated such that a more rigorous approach is used to aggregate fire detections, which better accounts for larger fires and enables using multiple satellite products simultaneously for emission estimates. Fuel characterization and emission factors have also been updated in FINNv2.5. Daily fire emissions for many trace gases and aerosols are determined for 2002-2019 (the Moderate Resolution Imaging Spectroradiometer (MODIS)-only fire detections) and 2012-2019 (MODIS+VIIRS fire detections). The non-methane organic gas emissions are allocated to the species of several commonly used chemical mechanisms. We compare FINNv2.5 emissions against other widely used fire emission inventories. The performance of FINNv2.5 emissions as inputs to a chemical

transport model is assessed with satellite observations. Uncertainties in the emission estimates remain, particularly Africa and South America during August – October, and in southeast and equatorial Asia in March and April. Recommendations for future evaluation and use are given.

## 1 Introduction

Open fires, such as wildfires, prescribed burns, agricultural fires, and land-clearing fires, are sources of atmospheric pollutants. Fire activity contributes to local, regional, and global emissions of greenhouse gases including carbon dioxide ($CO_2$) and methane ($CH_4$); reactive gases such as non-methane organic gases (NMOG), and nitrogen oxides ($NO_x$) that form ozone; dioxins and other air toxics, and particulate matter (PM). Fire emissions and their transport change atmospheric composition to cause impacts at many scales with implications for air quality (e.g., Burke et al., 2021; Jaffe et al., 2020; Liu et al., 2017; Tang et al., 2022; Bourgeois et al., 2021; Xu et al., 2020, Xu et al. 2021), regional and global climate (e.g., Dintwe et al., 2017; Jin et al., 2012; Liu et al., 2019), visibility (e.g., Ford et al., 2018; Jaffe et al., 2020; Val Martin et al., 2015), and human health outcomes (e.g., Liu et al., 2017; Reid et al., 2016; Xu et al., 2020). Many factors contribute to the spatial and temporal patterns and severity of fires and their emissions including agricultural, forest, and waste management practices; land use change; climatic factors such as temperature, rainfall, and drought conditions; and ecosystem diversity and health (e.g., Armenteras et al., 2021; Kelly et al., 2020; Pausas and Keeley, 2021). Future climate, policy, and human development patterns, including in the wildland-urban interface (WUI), will have complex interactions on the effects of fires that may require adaptive strategies for communities (Schoennagel et al., 2017).

Accurate estimates of fire emissions are required to understand chemistry and climate, to assess ambient pollutant concentrations and population exposure, and to evaluate the effectiveness of emissions control programs for air quality planning and management. The Fire INventory from NCAR (FINN) (Wiedinmyer et al., 2011, https://www2.acom.ucar.edu/modeling/finn-fire-inventory-ncar) was developed more than 10 years ago to provide daily global estimates of pollutant emissions from open fires with a high spatial and temporal resolution for use in air quality, atmospheric composition, and climate modeling applications. The National Center for Atmospheric Research (NCAR) has served as the central repository for FINN global emissions files spanning 2002-2020: http://bai.acom.ucar.edu/Data/fire/. FINN estimates have been downloaded more than 13450 times since 2013-08-29, as of 26 October 2022,

and the original model has been cited 908 times (Clarivate Web of Science). FINN emissions estimates have been applied in regions of the world that experience high fire activity to evaluate the influences on air quality and public health (e.g., Crippa et al., 2016; Kulkarni et al., 2020; Nawaz and Henze, 2020; Nuryanto, 2015; Pimonsree and Vongruang, 2018; Takami et al., 2020); to assess emissions trends (e.g., Ma et al., 2018; Shen et al., 2019); to examine the effects of changing climate and development patterns on wildfire emissions (e.g., Hurteau et al., 2014); and in comparisons with surface, aircraft and satellite-based observations (e.g., Reddington et al., 2019; Stavrakou et al., 2016), as well as with inventories developed using other fire emissions modeling systems (e.g., Bray et al., 2018; Faulstich et al., 2022; Kiely et al., 2019; Koplitz et al., 2018; Larkin et al., 2014; T. Liu et al., 2020; Pereira et al., 2016; Urbanski et al., 2018). Real-time emissions estimates from FINN version 1 (FINNv1) are currently used in the NCAR Whole Atmosphere Community Climate Model (WACCM) chemistry and aerosol forecasts (http://www.acom.ucar.edu/waccm/forecast/).

The FINNv1 model is based on a bottom-up approach to estimate emissions described by Wiedinmyer et al., (2011). In FINNv1, global observations from the Moderate Resolution Imaging Spectroradiometer (MODIS) sensors onboard the National Aeronautics and Space Administration's (NASA's) Terra and Aqua satellites are used to detect fire activity, beginning with the MODIS Rapid Response (MRR) system or the MODIS Data Processing System (MODAPS) Collection 5 (NASA/University of Maryland, 2002; Davies et al., 2009). Fuel characterization in FINNv1 is based on the Collection 5 MODIS Land Cover Type (LCT) product for 2005 (Friedl et al., 2010) with land cover classifications defined by the International Geosphere-Biosphere Programme (IGBP) and Collection 3 MODIS Vegetation Continuous Fields (VCF) product for 2001 (Carroll et al., 2011; Hansen et al., 2003; Hansen et al., 2005). Fuel loadings are assigned from Hoelzemann (2004) or Akagi et al. (2011). Estimates of fuel burned use the approach of Ito and Penner (2004). Emission factors by land cover classification for trace gases and particulate air pollutants in FINNv1 are based on published literature at the time (Akagi et al., 2011; Andreae and Merlet, 2001; Andreae and Rosenfeld, 2008; McMeeking, 2008).

FINN version 2.5 (FINNv2.5) has extensive updates to the input data and processing used for the detection of fire activity, characterization of annual land use/land cover and vegetation density, determination of area burned, and the application of fuel loadings by global region compared to the FINNv1 configuration. FINNv2.5 also includes revisions to emission factors based on current literature. Here we describe the development of FINNv2.5, released in 2022 (Wiedinmyer and Emmons, 2022). Global emissions for

2002 through 2021 have been created and are online for public use, for inclusion in emissions inventories and chemical and climate modeling applications, and for comparisons with previous versions of FINN and results from other fire emissions models.

## 2 Methods

FINNv2.5 uses the same FINNv1 bottom-up methodology (Seiler and Crutzen, 1980; Wiedinmyer et al., 2011) as defined by the overall equation:

$$E_i = A(x,t) \times B(x) \times FB \times EF_i \qquad (1)$$

where the emissions (E, mass of pollutant i) is the product of the area burned at location $x$ and time $t$ [A(x,t)], the biomass at location $x$ [B(x)], the fraction of biomass that is burned (FB), and an emission factor (EF$_i$, mass of pollutant $i$ per biomass burned).

The FINNv2.5 model framework has three components: (1) burned area and land cover determination (Sections 2.1-2.3), (2) fuel consumption and emission calculation (Section 2.4), and (3) speciation of the non-methane organic gases (NMOG) (Section 2.5).

### 2.1 Fire location and timing

FINNv2.5 first determines burned area from daily satellite detections of active fires. FINNv2.5 uses MODIS detections (nominal 1 km$^2$ resolution) (Giglio et al., 2006), as in FINNv1, and adds the option to use active fire detections at 375 m resolution from the Visible Infrared Imaging Radiometer Suite (VIIRS; Csiszar et al., 2014), onboard the Suomi National Polar-orbiting Partnership (Suomi-NPP) satellite, alone or in combination with MODIS active fire data. The use of VIIRS 375m detections is a major advancement from the use of MODIS-only fire detections, as this product better captures small fires. VIIRS detections are available from 2012 to the current year.

The MODIS Collection 6 (MCD14DL) and VIIRS active fire products are obtained from NASA's Fire Information for Resource Management System (FIRMS) data portal: https://firms.modaps.eosdis.nasa.gov/download/. The MODIS product provides the location, overpass

time (Coordinated Universal Time, UTC), and confidence of daily fire detections. Data confidence in the MODIS product is specified by a numeric scale of 0 to 100%. Detections with a confidence specification of less than 20% are eliminated from our calculation in FINNv2.5, as was done in earlier FINN versions. Daily global coverage is not accomplished at latitudes between approximately 23.5° N and 23.5°S due to the observational swath width. To account for the lack of daily observations, fire detections only in these equatorial regions are counted for a two-day period: each fire is assumed to continue into the next day. Regardless of whether the detection is from MODIS or VIIRS, the intent is to repeat the fire at the same location since there is not a clear alternative at this time in these global regions, as described by Wiedinmyer et al. 2011.

With its improved spatial resolution of 375m, the VIIRS product provides more sensitive detection of fires of relatively small areas, fully global coverage, improved mapping of large fire perimeters, and improved nighttime performance relative to MODIS fire detections (Schroeder et al., 2014). The higher detection rates of small fires can be particularly important for areas of the world such as southeast Asia, where agriculture burning is common, and in the southeast U.S., where there is a large amount of managed burning. Detection confidence is provided by the VIIRS product and is specified by three categories, low/nominal/high. In the FINNv2.5 preprocessor, detections with a confidence specification identified as "low" are eliminated from the analysis. We only include data attributed to thermal anomalies from vegetation fires (Type=0), i.e., other thermal anomaly types associated with active volcanos or other static land sources are filtered from the product.

The processing of the two simultaneous fire products in FINNv2.5 does not lead to double counting fires: the FINNv2.5 method determines the spatial union of all adjacent detections for a given day as the daily burned area of a fire, as described below in Section 2.2. The identity of the sensor is not relevant for the determination of burned area, as long as the pixel size for each detection is correctly represented (i.e., $0.14 \text{ km}^2$ for VIIRS and $1 \text{ km}^2$ for MODIS). FINN v2.5 and earlier versions do not account for the effects of burning from earlier days for fire detections at the same location, i.e., the fraction burned is consistent

with unperturbed vegetation on the first day regardless of the persistence of fire activity, and emissions scale directly with the detection of a thermal anomaly in the same location over multiple days.

The active fire products report the time of acquisition in Coordinated Universal Time (UTC). In contrast to previous versions of the model, the FINNv2.5 preprocessor uses local time in the specification of the date of a fire detection in order to facilitate comparisons of emissions estimates with observational data:

$$\text{Local Time} = \text{UTC} + \text{Nearest\_Integer (Longitude/15)} \tag{2}$$

## 2.2 Burned area

FINNv1 estimates burned area for each fire pixel identified individually, and the nominal pixel size for the MODIS fire detections, 1 $km^2$, is assumed per detection. Spatially overlapping detections are eliminated from further analysis. It was recognized that for large fires in forested regions, an array of multiple discrete detections is typically reported, and an estimate of a contiguous area that represents the total area burned by a fire is needed. We improved the burned area estimate in FINNv2 to better represent the area associated with each fire.

A fire event in the western United States is shown in Figure 1a to illustrate the new approach for estimating area burned. For FINNv2.5, each reported active fire detection (Figure 1b) is assigned a square area of 0.14 $km^2$ from VIIRS or 1 $km^2$ from MODIS (Figure 1c) based on the nominal horizontal resolution of the data ("instrument resolution square"). Detections determined to be in proximity with one another are aggregated by two different approaches, depending on the land cover type and forest cover. Initially, it is assumed that multiple detections by adjacent pixels in a satellite sensor array are part of a larger fire, and these detections are merged. The scan and track sizes of the satellite pixel are provided by the fire detection product and define the actual resolution of the fire detection. The scan and track sizes for each fire detection are used for identifying groups of records that represent contiguous or overlapping detections. A rectangle with easterly and northerly sizes equal to 110% of the scan and track sizes is established for each detection ("detection rectangle") (Figure 1d), with the objective of identifying adjacent neighboring detections, but not for direct application to the burned area estimation. Fire detections are identified as being from one larger fire when any of the satellite detection rectangles

overlap. To minimize an overestimation of burned area, a convex hull is generated between corresponding pairs of detection rectangles that directly intersect. The union of pairwise convex hulls from a cluster forms an "extended fire polygon" that represents the tentative estimated burned area for a single fire event or group of nearby fires for the day (Figure 1d). This approach effectively fills any gap between instrument

resolution squares.

For each of the extended polygons, the MOD44B v006 MODIS/Terra VCF annual product (https://lpdaac.usgs.gov/products/mod44bv006/) is overlaid (Section 2.3), and average tree cover fraction is determined (Figure 1e). For forested areas with tree cover $\geq$ 50% as determined by the VCF product, the merged polygons are accepted as the final burned area estimate. Otherwise, the merging is not used,

and instead an alternative, more conservative approach is applied to determine the burned area for the region. This alternative approach is used to prevent overestimation of emissions in regions with many small fires, as in the savanna fires in sub-Saharan Africa. The alternative polygon aggregation is achieved by aggregating nearby detections only when the instrument pixels themselves are intersecting (Figure 1c) and therefore not with the extended detection footprints (Figure 1d). The result is an aggregation

algorithm that is repeated with a smaller set of detections to determine the alternative conservative set of polygons ("conservative fire polygon"). The "final burned area polygons" in Figure 1f show examples of a composite of polygons based on these two different aggregation approaches. Note at the bottom of Figure 1f, a region with less than 50% tree cover has a smaller final burned area polygon estimate than the extended polygon (blue). In contrast, the polygon in the center of Figure 1f, which is a forested area

with more than 50% tree cover uses the extended polygon determined in Figure 1d. (Further information

about this aggregation is in Section S1 of the Supporting Information).

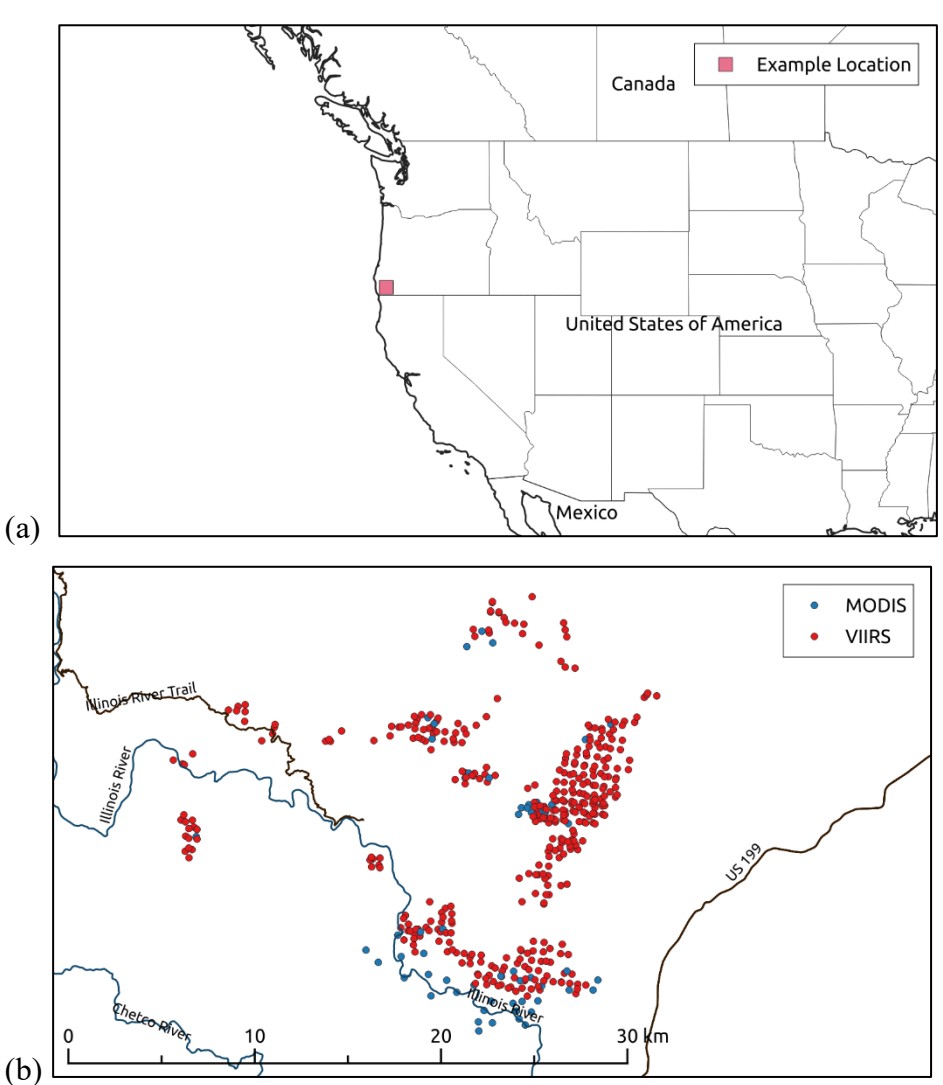

(a)

(b)

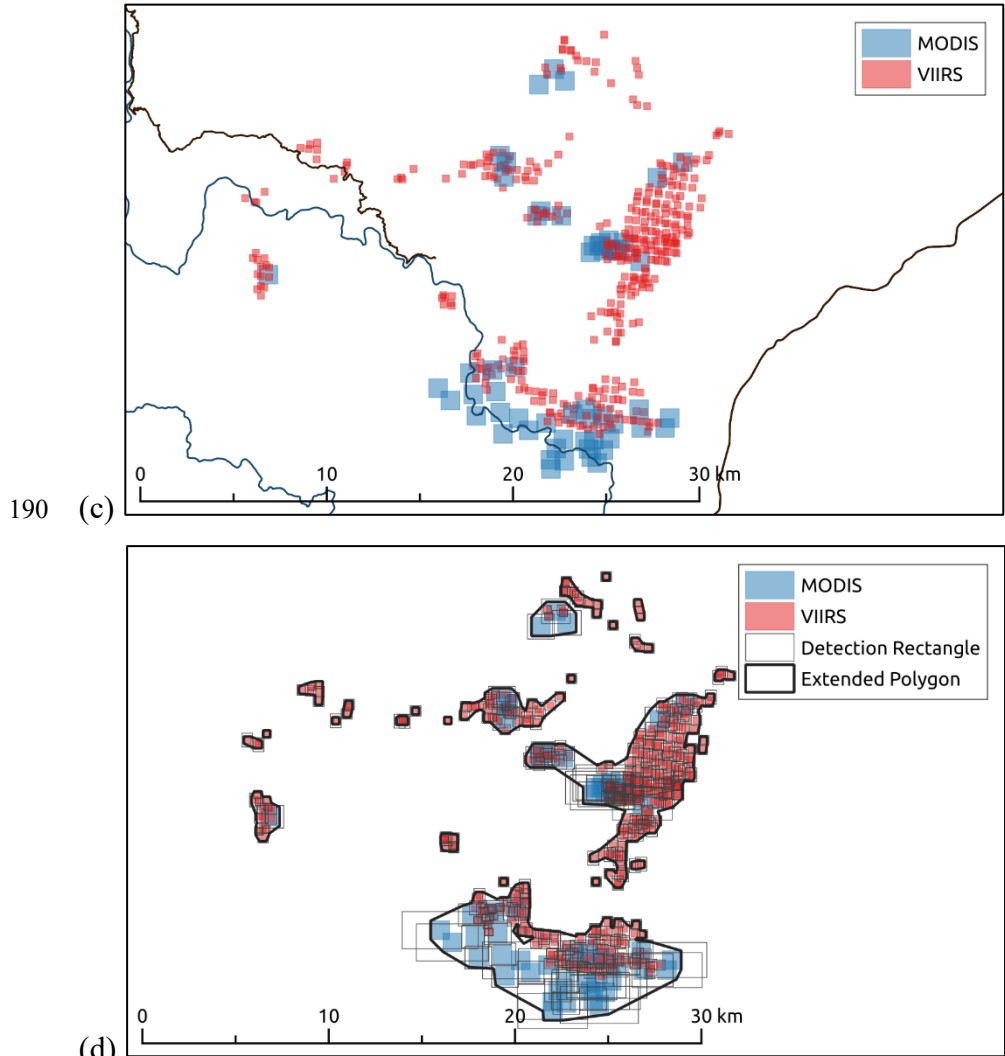

(c)

(d)

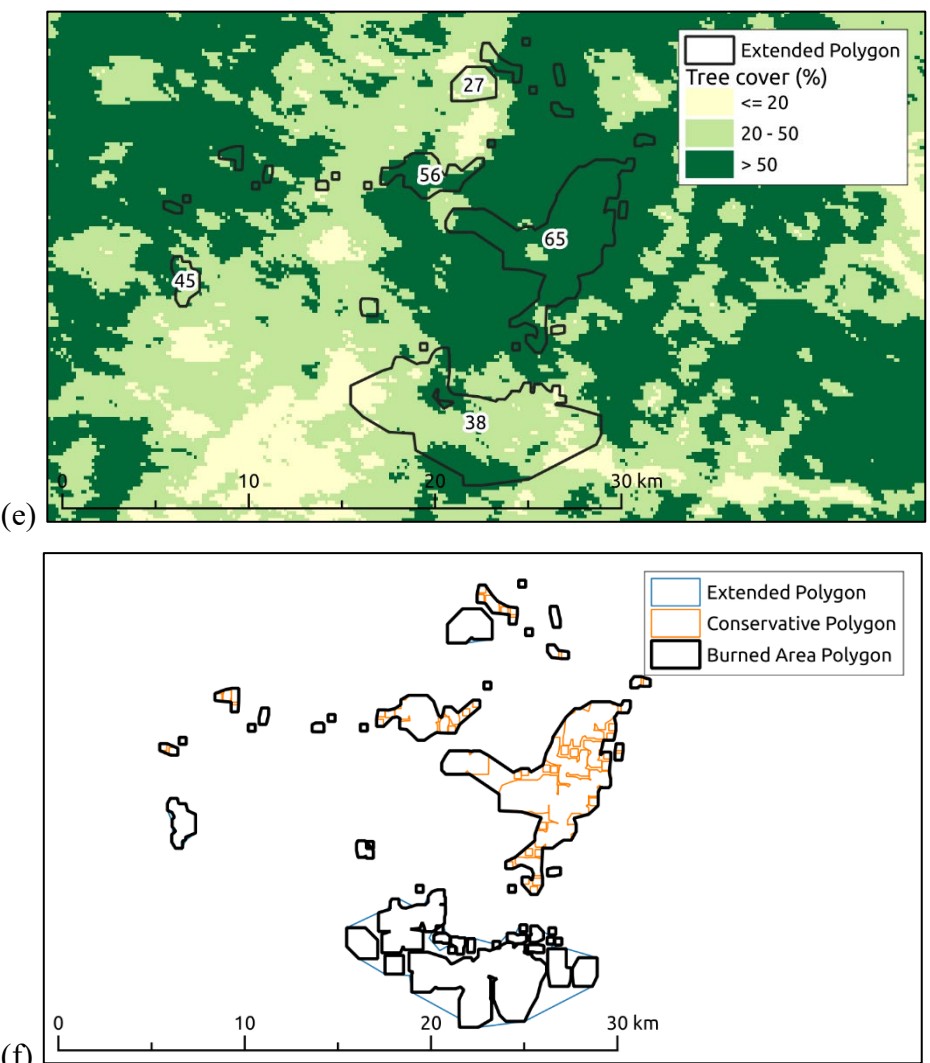

(e)

(f)

**Figure 1. Illustration of the burned area determination used in the FINNv2.5 preprocessor for 11 August 2018 when both MODIS and VIIRS fire detections are applied: (a) example fire location; (b) active fire detections; (c) burned area per detection based on instrument resolution; (d) detection clusters joined for determination of extended burned area; (e) determination of average percent tree cover (labelled as VCF) for merged polygons; (f) final burned area polygons reflecting either extended or conservative polygons based on percent tree cover.**

Subsequently, the final burned area polygons are subdivided using a Voronoi tessellation algorithm in order to develop emissions estimates by land cover classification as described in Sections 2.3 and 2.4.

Each of the undivided, final burned area polygons are assigned a unique fire id to facilitate users to group emission estimates from a presumed single fire event.

## 2.3 Fuel loading and vegetation inputs

The NASA MODIS VCF product provides estimates of the percentage of bare surface, herbaceous, and
forested cover at a horizontal resolution of 250m (Figure S1). For each fire area, the subdivided polygons described in Section 2.2 (Figure 1g) are overlaid on the vegetation cover data from MOD44B v006 MODIS/Terra VCF annual product (https://lpdaac.usgs.gov/products/mod44bv006/) (Figure 1e). The VCF data for the prior year are chosen, so that the VCF before any land cover changes due to fire are used in the emission estimation process. The VCF raster is clipped to the geometry of the fire polygon,
and the averages of the VCF tree, herbaceous and bare cover are calculated for each fire polygon.

FINNv2.5 uses the Terra and Aqua combined MODIS LCT MCD12Q1 Version 6 data product with the International Geosphere-Biosphere Programme (IGBP) classification scheme (https://lpdaac.usgs.gov/products/mcd12q1v006/) as its default land cover information. Figure 2 shows the global distribution of land cover applied as the default in FINNv2.5. Each subdivided polygon (for example, in Figure 3) is assigned factional coverage of one of 16 land cover classifications (Table 1). Similar to the application of the VCF information, land use data from the previous year are used.

Use of the LCT and VCF products in FINNv2.5 is an improvement from FINNv1. FINNv1 used one static map of LCT and VCF (from 2002) for any year processed. FINNv2.5 employs year-specific MODIS LCT and VCF maps that change annually. Further, the specific vegetation assignments for each subdivided polygon enables different vegetation types and coverage to be represented across larger fires. These input data and processes enable better representation of the vegetation that is burned.

All fire polygons are assigned to one of 13 global regions (Wiedinmyer et al. 2011) used to assign fuel loadings (Section 2.4). This completes the first component of the FINNv2.5 modeling framework and results in a file of daily burned areas and associated land cover information.

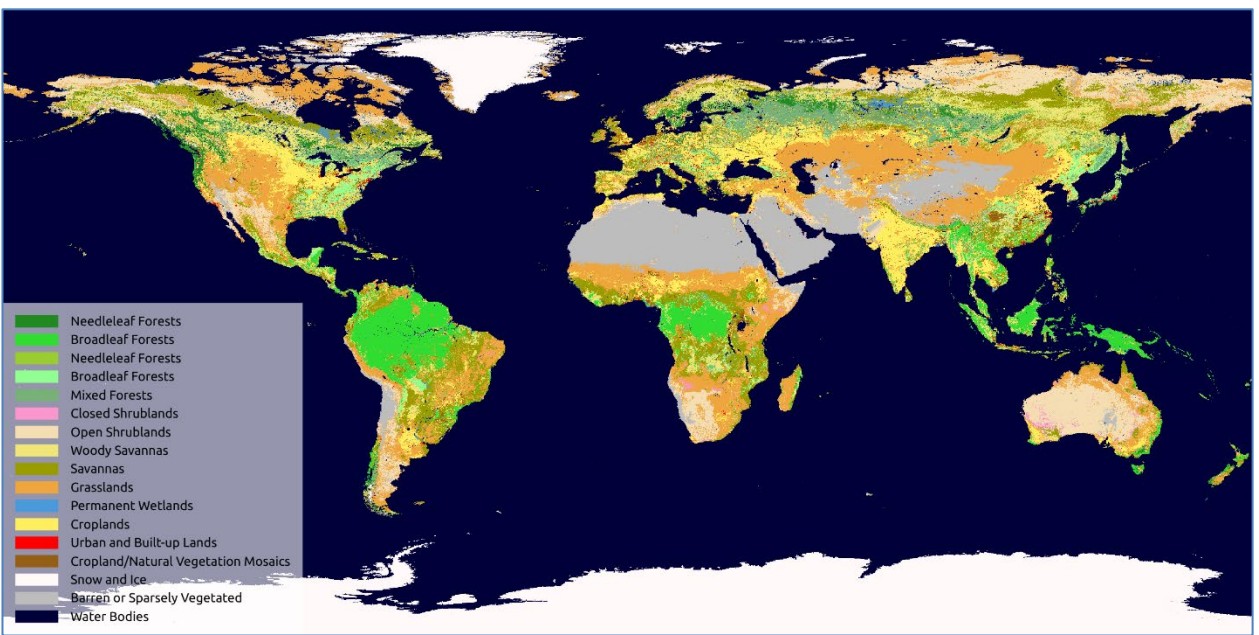

**Figure 2: MCD12Q1 Version 6 data product with the International Geosphere-Biosphere Programme (IGBP) classification scheme.**

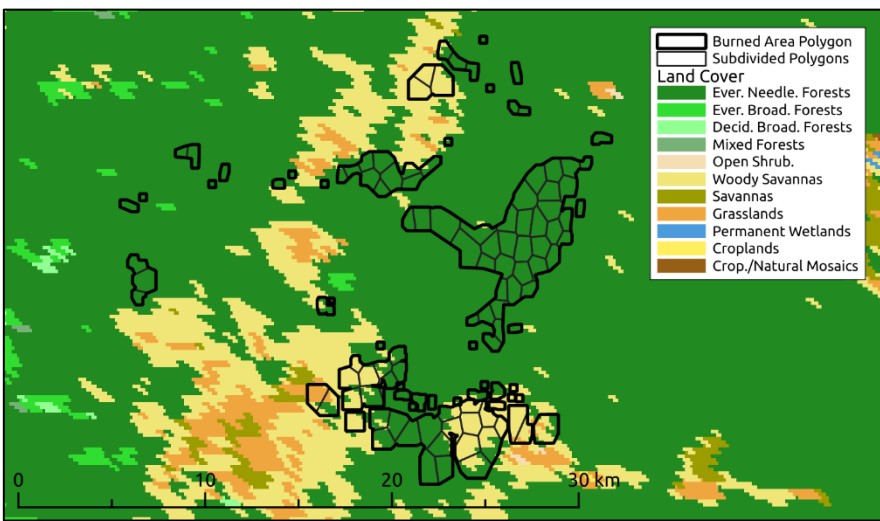

**Figure 3: Illustration of the burned area determination used in the FINNv2.5 preprocessor for 11 August 2018 continued from Figure 1: subdivision of burned area polygons to develop emissions estimates by the MODIS IGBP land cover classification.**

## 2.4 Emissions Calculations

The next step of the model framework is the emissions calculation. In this step, the daily burned area and associated vegetation information (described above) are assigned associated fuel loadings. Using the same process described by Wiedinmyer et al. (2011), where the biomass burned is assigned based on land cover type and global region (B), the fraction of the biomass that is burned (FB) is assigned as a function of tree and herbaceous cover, emission factors (EF) are determined based on land cover, and daily pollutant emissions estimates are calculated following equation 1. Overall, the emission calculation process follows this framework as described by Wiedinmyer et al. (2011), with the following exceptions.

Similar to earlier FINN versions (Wiedinmyer et al., 2011), the 16 IGBP land cover classifications of the LCT product are mapped to consolidated vegetation types, depending on the land cover class and latitude that distinguish tropical, temperate and boreal forests (Table 1). The consolidated vegetation types used in FINNv2.5 are grassland and savanna, woody savanna or shrubs, tropical forest, temperate forest, boreal forest, temperate evergreen forest, and crops.


**Table 1. LCT IGBP and generic vegetation type descriptions.**

| IGBP LCT Description | LCT Value | Generic Vegetation Type | Genveg Value |
|---|---|---|---|
| Evergreen Needleleaf Forests | 1 | if latitude > 50, then Boreal Forest; else Temperate Evergreen Forest | 5,6 |
| Evergreen Broadleaf Forests | 2 | if latitude > -23.5 and < 23.5, then Tropical Forest; else Temperate Forest | 3, 4 |
| Deciduous Needleleaf Forests | 3 | if latitude > 50, then Boreal Forest; else Temperate Forest | 5,4 |
| Deciduous Broadleaf Forests | 4 | Temperate Forest | 4 |
| Mixed Forests | 5 | If latitude > 5, then Boreal Forest; if latitude > -23.5 and < 23.5 then Tropical Forest; else Temperate Forest | 5, 3, 4 |
| Closed Shrublands | 6 | Woody Savanna or Shrubs | 2 |
| Open Shrublands | 7 | Woody Savanna or Shrubs | 2 |
| Woody Savannas | 8 | Woody Savanna or Shrubs | 2 |
| Savannas | 9 | Grassland and Savanna | 1 |
| Grasslands | 10 | Grassland and Savanna | 1 |
| Permanent Wetlands | 11 | Grassland and Savanna | 1 |
| Croplands | 12 | Croplands | 9 |
| Urban and Built-up Lands | 13 | If Treecover < 40, then reassign to 10; If Treecover > 40 and < 60, then reassign to 8; If tree > 60 then assign based on latitude | ** |
| Cropland/Natural Vegetation Mosaics | 14 | Grassland and Savanna | 1 |
| Permanent Snow and Ice | 15 | Remove | |
| Barren | 16 | Grassland and Savanna | 1 |
| Water Bodies | 17 | Remove | |
| Unclassified | 255 | Remove | |

*\*\* if latitude > 50, then Boreal Forest; if latitude > -30 and < 30, then Tropical Forest; Else, Temperate Forest*

The fuel loading, or the potential maximum amount of biomass available to be burned [B(x) in equation
1], is assigned by generic vegetation type and global region (Table 2). Selected values were updated for
FINNv2.5 from earlier versions of FINN based on van Leeuwen et al. (2014). The fuel loading for crops
was updated to 902 g m$^{-2}$ based on an average from (Akagi et al., 2011; van Leeuwen et al., 2014; Pouliot
et al., 2017). Specific crop types are not identified in the version described here.


**Table 2. Fuel loadings (g m⁻²) assigned by generic land cover type and global region. These values are as described by Wiedinmyer et al. (2011) unless noted otherwise. Highlighted values indicate those updated for FINNv.2.5 based on van Leeuwen et al. (2014).**

| Global Region | Tropical Forest | Temperate Forest | Boreal Forest | Woody Savanna/ Shrublands | Savanna and Grasslands[g] |
|---|---|---|---|---|---|
| North America | 28,076 | 10,661[e] | 17,875[e] | 4,762 | 976 |
| Central America | 26,500[e] | 11,000 | | 2,224 | 418 |
| South America | 26,755[e] | 7,400 | | 3,077 | 624[e] |
| Northern Africa | 25,366 | 3,497 | | 2,501 | 382[e] |
| Southern Africa | 25,295 | 6,100 | | 2,483 | 411[e] |
| Western Europe | 28,076 | 7,120 | 6,228 | 4,523 | 1,321 |
| Eastern Europe | 28,076 | 11,386 | 8,146 | 7,752 | 1,612 |
| North Central Asia | 6,181 | 20,807 | 14,925[e] | 11,009 | 2,170 |
| Near East | 6,181 | 10,316 | | 2,946 | 655 |
| East Asia | 14,941[e] | 7,865 | | 4,292 | 722 |
| Southern Asia | 26,546[e] | 14,629 | | 5,028 | 1,445 |
| Oceania | 16,376 | 13,535[e] | | 2,483[f] | 552[e] |

[a] Akagi et al. (2011) and references therein; [b] tropical forest class added for North America and Europe (in LCT); [c] all Asia assigned equal tropical forest values; [d] taken as the average of tropical and temperate forest fuel loadings for Oceania; [e] (van Leeuwen et al., 2014); [f] taken as the same for African woody savanna from van Leeuwen et al., 2014; [g] croplands assigned same fuel loading as grasslands.


For North America, FINNv2.5 utilizes the fuel loadings for coarse/woody and herbaceous vegetation by land cover type derived from the Fuel Characteristic Classification System (FCCS) of the U.S. Department of Agriculture Forest Service (https://www.fs.fed.us/pnw/fera/fccs/), as described by McDonald-Buller et al. (2015). These fuel loadings (Table 3) have priority over the regional default fuel loadings (Table 2).

**Table 3. North American fuel loadings (g m⁻²) by land cover type for coarse/woody and herbaceous vegetation. These values are based on the Fuel Characteristic Classification System (https://www.fs.fed.us/pnw/fera/fccs/) with the exception of croplands[a].**

| Land Cover Type | Fuel Loading $(g\ m^{-2})$ | |
|---|---|---|
| | Coarse/Woody | Herbaceous |
| Water | 0 | 0 |
| Evergreen Needleleaf Forest | 28,930 | 437 |
| Evergreen Broadleaf Forest | 19,917 | 650 |
| Deciduous Needleleaf Forest | 15,653 | 541 |
| Deciduous Broadleaf Forest | 19,982 | 964 |
| Mixed Forests | 20,339 | 766 |
| Closed Shrublands | 5,136 | 229 |
| Open Shrublands | 2,889 | 169 |
| Woody Savannas | 12,907 | 668 |
| Savannas | 10,907 | 764 |
| Grasslands | 2,822 | 407 |
| Permanent Wetlands | 8,509 | 712 |
| Croplands | 0 | 902[a] |
| Urban and Built-Up | 0 | 0 |
| Cropland/Natural Vegetation | 9,080 | 822 |
| Snow and Ice | 0 | 0 |
| Barren or Sparsely Vegetated | 1,355 | 104 |

aTaken as an average from van Leeuwen et al. (2014), Akagi et al., (2011), and McCarty et al. (2012).

Emission factors are assigned based on the generic vegetation type. Since the original release of FINNv1 in 2011, there have been many studies to measure emission factors from wildland fires. We have updated the emission factors from FINNv1.5 with results from recent publications (Table 4).

**Table 4: Emission factors (g kg$^{-1}$) for FINNv2.5.**

| Chemical Species | Generic Vegetation Index and Type | | | | | | |
|---|---|---|---|---|---|---|---|
| | 1 | 2 | 3 | 4 | 5 | 6 | 9 |
| | Savanna Grasslands[1] | Woody Savanna/ Shrubs | Tropical Forest | Temperate Forest[2] | Boreal[3] | Temperate Evergreen Forest[2] | Crops[4] |
| Carbon Dioxide (CO2) | 1686 | 1681 | 1643 | 1510 | 1565 | 1623 | 1444 |
| Carbon Monoxide (CO) | 63 | 67 | 93 | 122 | 111 | 112 | 91 |
| Methane (CH4) | 2 | 3 | 5.1 | 5.61 | 6 | 3.4 | 5.82 |
| Non-methane Organic Gases (NMOG)[5] | 28.2 | 24.8 | 51.9 | 56 | 48.5 | 49.3 | 51.4 |
| Hydrogen (H2) | 1.7 | 0.97 | 3.4 | 2.03 | 2.3 | 2 | 2.59 |
| Nitrogen Oxides (NOXasNO) | 3.9 | 3.65 | 2.6 | 1.04 | 0.95 | 1.96 | 2.43 |
| Sulfur Dioxide (SO2) | 0.9 | 0.68 | 0.4 | 1.1 | 1 | 1.1 | 0.4 |
| Particulate Matter with Diameters less than 2.5 μm (PM25) | 7.17 | 7.1 | 9.9 | 15 | 18.4 | 17.9 | 6.43 |
| Total Particulate Matter (TPM) | 8.3 | 15.4 | 18.5 | 18 | 18.4 | 18 | 13 |
| Total Particulate Carbon (TPC) | 3 | 7.1 | 5.2 | 9.7 | 8.3 | 9.7 | 4 |
| Particulate Organic Carbon (OC) | 2.6 | 3.7 | 4.7 | 7.6 | 7.8 | 7.6 | 2.66 |
| Particulate Black Carbon (BC) | 0.37 | 1.31 | 0.52 | 0.56 | 0.2 | 0.56 | 0.51 |
| Ammonia (NH3) | 0.56 | 1.2 | 1.3 | 2.47 | 1.8 | 1.17 | 2.12 |
| Nitrogen Oxide (NO) | 2.16 | 0.77 | 0.9 | 0.95 | 0.83 | 0.95 | 1.18 |
| Nitrogen Dioxide (NO2) | 3.22 | 2.58 | 3.6 | 2.34 | 0.63 | 2.34 | 2.99 |
| Non-methae Hydrocarbons (NMHC) | 3.4 | 3.4 | 1.7 | 5.7 | 5.7 | 5.7 | 7 |
| Particulate Matter with Diameters less than 10 μm (PM10) | 7.2 | 11.4 | 18.5 | 16.97 | 18.4 | 18.4 | 7.02 |

[1]Emission factors for Tropical Forests, Savannah/Grasslands, Woody Savannah/Shrubs updated to average values from Akagi et al. (2011) (updated Feb. 2015). [2]Emission factors for Temperate Forest and Temperate Evergreen Forests are average values from Akagi et al. (2011) (updated Feb 2015) and results from X. Liu et al. (2017), Paton-Walsh et al. (2014), and Urbanski (2014). For Temperate Evergreen Forest, only results from evergreen forests included. [3]Boreal Forest emission factors are average of Akagi et al. (2011) with emission factors from boreal forest emission factors from Urbanski et al. (2014).[4]Crop

emission factors updated with average values from Akagi et al (2011) and results from Fang et al. (2017), X. Liu et al. (2016), Santiago-De La Rosa et al. (2018), and Stockwell et al. (2015) Table S3). [5]NMOG emission factors now include identified and unidentified compounds.

## 2.5 Allocation of Non-Methane Organic Gases to Chemical Mechanisms

Wiedinmyer et al (2011) provided mappings from the total mass NMOG emission values calculated by
the FINN model to the surrogate species of three chemical mechanisms commonly used in chemical
transport models: MOZART, SAPRC99, and GEOS-Chem. The mapping of NMOG emissions to the
MOZART T1 chemical mechanism was created for FINNv2.5 based on recent published emissions data
and updates in the chemical mechanisms (Table 5). To apply, the total NMOG mass estimated by FINN
should be multiplied by the mapping value (mole species kg NMOG$^{-1}$, e.g., Table 5) to assign the molar
emission of the surrogate species. The mappings to the SAPRC99 and GEOS-chem have not been updated
and are the same as described by (Wiedinmyer et al., 2011).

Table 5: Factors to map the total NMOG mass emissions to the MOZART-T1 chemical species (mole-species kgNMOG$^{-1}$).

| MOZART T1 Chemical Species | Generic Vegetation Index and Type | | | | | | |
|---|---|---|---|---|---|---|---|
| | 1 | 2 | 3 | 4 | 5 | 6 | 9 |
| | Savanna Grasslands | Woody Savanna/ Shrubland | Tropical Forest | Temperate Forest | Boreal Forest | Temperate Evergreen Forest | Crop |
| APIN | 0.009 | 0.053 | 0.0 | 0.261 | 0.259 | 0.261 | 0.010 |
| BENZENE | 0.144 | 0.442 | 0.0 | 0.253 | 0.290 | 0.253 | 0.091 |
| BIGALK | 0.156 | 0.644 | 0.219 | 0.415 | 1.821 | 0.415 | 0.246 |
| BIGENE | 1.467 | 1.274 | 0.662 | 1.393 | 0.627 | 1.393 | 0.674 |
| BPIN | 0.0 | 0.004 | 0.0 | 0.008 | 0.209 | 0.008 | 0.0 |
| BZALD | 0.791 | 0.272 | 0.120 | 0.298 | 0.166 | 0.298 | 0.325 |
| C2H2 | 2.103 | 1.975 | 0.672 | 2.513 | 1.167 | 2.513 | 1.701 |
| C2H4 | 1.218 | 2.886 | 1.505 | 1.930 | 1.407 | 1.930 | 1.412 |
| C2H6 | 0.859 | 0.641 | 0.939 | 0.611 | 1.168 | 0.611 | 0.673 |
| C3H6 | 0.647 | 0.557 | 0.603 | 0.487 | 0.499 | 0.487 | 0.457 |
| C3H8 | 0.090 | 0.561 | 0.114 | 0.149 | 0.194 | 0.149 | 0.142 |
| CH2O | 1.532 | 2.285 | 2.299 | 2.181 | 1.361 | 2.181 | 1.716 |
| CH3CH2OH | 0.0 | 0.055 | 0.0 | 0.066 | 0.023 | 0.066 | 0.0 |
| CH3CHO | 1.037 | 0.792 | 1.404 | 0.758 | 0.416 | 0.758 | 0.929 |
| CH3CN | 0.117 | 0.130 | 0.399 | 0.088 | 0.176 | 0.088 | 0.142 |
| CH3COCH3 | 0.201 | 0.242 | 0.433 | 0.297 | 0.242 | 0.297 | 0.162 |
| CH3COOH | 2.371 | 1.353 | 2.029 | 1.292 | 1.360 | 1.292 | 2.349 |
| CH3OH | 1.451 | 1.650 | 3.031 | 1.744 | 1.608 | 1.744 | 2.328 |

| | | | | | | | |
|---|---|---|---|---|---|---|---|
| CRESOL | 0.059 | 0.058 | 0.0 | 0.059 | 0.040 | 0.059 | 0.074 |
| GLYALD | 0.390 | 0.128 | 1.886 | 0.210 | 0.233 | 0.210 | 0.800 |
| HCN | 0.559 | 0.927 | 0.625 | 0.684 | 0.846 | 0.684 | 0.416 |
| HCOOH | 0.206 | 0.134 | 0.683 | 0.259 | 0.254 | 0.259 | 0.426 |
| HONO | 0.298 | 0.643 | 1.001 | 0.326 | 0.228 | 0.326 | 0.187 |
| HYAC | 0.309 | 0.118 | 0.609 | 0.223 | 0.149 | 0.223 | 1.548 |
| ISOP | 0.069 | 0.138 | 0.029 | 0.129 | 0.085 | 0.129 | 0.062 |
| LIMON | 0.0 | 0.013 | 0.0 | 0.158 | 0.0 | 0.158 | 0.0 |
| MACR | 0.0 | 0.147 | 0.222 | 0.113 | 0.024 | 0.113 | 0.0 |
| MEK | 0.370 | 0.286 | 0.666 | 0.274 | 0.104 | 0.274 | 0.387 |
| MGLY | 0.347 | 0.094 | 0.0 | 0.135 | 0.090 | 0.135 | 0.171 |
| MVK | 0.317 | 0.301 | 0.222 | 0.247 | 0.087 | 0.247 | 0.193 |
| MYRC | 0.0 | 0.003 | 0.0 | 0.002 | 0.0 | 0.002 | 0.0 |
| PHENOL | 0.472 | 0.457 | 0.191 | 0.345 | 0.517 | 0.345 | 0.408 |
| TOLUENE | 0.457 | 0.531 | 0.769 | 0.605 | 1.327 | 0.605 | 0.375 |
| XYLENE | 0.385 | 0.355 | 0.040 | 0.422 | 0.238 | 0.422 | 0.295 |
| XYLOL | 0.108 | 0.046 | 0.0 | 0.088 | 0.056 | 0.088 | 0.130 |

## 3 Results

### 3.1 Emission estimates

The FINNv2.5 model was run in two ways to produce emissions for evaluation and assessment: (1) for comparison with the previous version of FINN (FINNv1.5) using MODIS-only fire detections and calculated starting in 2002 (FINNv2.5(MODIS)) and (2) using both MODIS and VIIRS fire detections and calculated starting in 2012 (FINNv2.5(MODIS+VIIRS). The FINNv2.5 emissions files are freely available for use by the community (Wiedinmyer and Emmons, 2022). Results through 2019 are

presented in this manuscript.

Estimates from several versions of FINN are compared to other emission inventories: Global Fire Emissions Database (GFED) (van der Werf et al., 2017), Fire Energetics and Emissions Research (FEER) (Ichoku & Ellison, 2014), the Global Fire Assimilation System (GFAS)(Kaiser et al., 2012), and the Quick

Fire Emissions Dataset (QFED) version 2.5 (Darmenov and da Silva, 2015)  (Figure 4). The comparisons
are done by global region that follows (Giglio et al., 2010) (Figure 4).

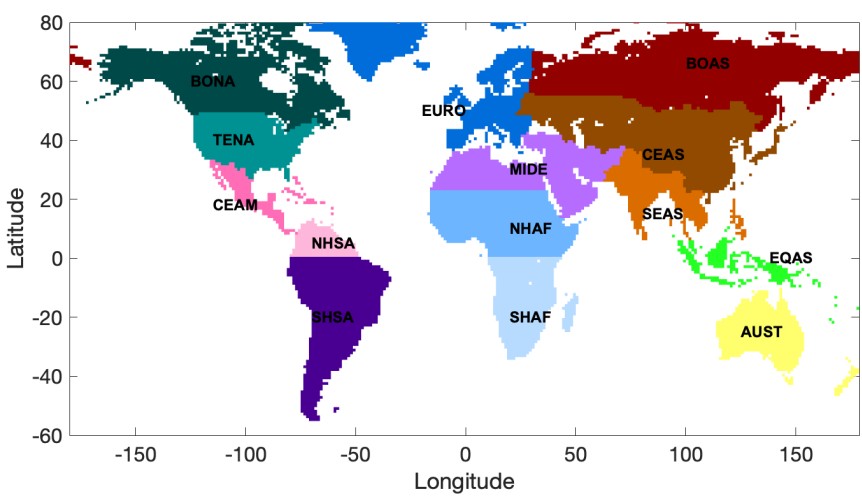

**Figure 4. Annually-averaged (2012-2019) emissions of CO₂, CO, formaldehyde (CH₂O), particulate black carbon (BC) + organic**

carbon (OC), ammonia ($NH_3$), ethane ($C_2H_6$), sulfur dioxide ($SO_2$), and nitrogen oxides ($NO_x$) from the Fire Inventory from NCAR version 2.5 (FINNv2.5(MODIS+VIIRS)) , FINNv2.5MODIS-only version (FINNv2.5 (MODIS)), FINNv1.5, Global Fire Emissions Database (GFED), Fire Energetics and Emissions Research (FEER), Global Fire Assimilation System (GFAS), and Quick Fire Emissions Dataset (QFED). Bars show global totals broken up into regional totals by color (Giglio et al., 2010), and shown in the global map here, namely Boreal North America (BONA), Temperate North America (TENA), Central America (CEAM), Northern Hemisphere South America (NHSA), Southern Hemisphere South America (SHSA), Europe (EURO), Middle East (MIDE), Northern Hemisphere Africa (NHAF), Southern Hemisphere Africa (SHAF), Boreal Asia (BOAS), Central Asia (CEAS), Southeast Asia (SEAS), Equatorial Asia (EQAS), and Australia and New Zealand (AUST). For comparisons with earlier versions of FINN, see Figure S2.

Figure 4 compares annual averaged (2012-2019) emissions of key pollutants from several versions of FINN and the other inventories by region. For all emitted species, FINNv2.5(MODIS+VIIRS) global emissions are higher than, and approximately double, those predicted by FINNv1.5. This is the case, even when only MODIS fire detections are considered. The increase in emissions from previous versions is primarily due to the new processing of area burned. In previous versions, the fire area was determined from a satellite detection pixel only; the updated version here also includes the composite of many detections into larger areas of fire activity (Section 2.1). The inclusion of VIIRS into the FINNv2.5(MODIS+VIIRS) inventory globally adds approximately 25% above the FINNv2.5(MODIS) processing for all emitted species. Further, emissions of NMOG and the individual species that make up NMOG (eg., CH2O and C2H6 in Figure 4) are increased significantly due to the use of updated emissions factors from recent field campaigns. Previous studies have shown low biases in FINN regional and species-specific estimates: for example, CO in the western US (Pfister et al., 2011) and Australia (Desservettaz et al., 2022), and particulate carbon in North America (Carter et al., 2020). The updated version is expected to correct some of these prior biases.

FINNv2.5(MODIS+VIIRS) emission estimates are overall at the higher end of the range of annual global total emissions provided by our sample of other commonly used emission inventories, likely due to a combination of the aggregated burned areas and the fact that FINNv2.5(MODIS+VIIRS) includes fire information from VIIRS, which captures more small fires. However, depending on the pollutant emitted, a comparison across different emission inventories shows varied results. For example, both $CO_2$ and CO global annual emissions from FINNv2.5 (MODIS and MODIS+VIIRS) are higher than QFED; but Black Carbon (BC) and ammonia ($NH_3$) are lower in FINNv2.5 than QFED. The primary drivers of these

differences are the assumed fuel type burned and associated emission factors. This difference in emission amounts between inventories is more variable when looking regionally and year to year (e.g. Figure 5 and S4).

In general, the year-to-year variabilities of annual fire emissions is consistent between different inventories, and mainly the magnitudes of emissions differ (Figures 5, S5, S6). FINNv2.5(MODIS+VIIRS) is often among the inventories that produce the highest CO emissions in all 14 global regions. Some notable exceptions are Boreal North America and Boreal Asia, where GFAS and sometimes GFED estimate higher emissions. This is likely due to the representation of smoldering peat fires in the high latitudes that are represented as a specific vegetation type in the GFED and GFAS inventories, but not in FINNv2.5. [Note: Kiely et al. (2019) developed a parameterization used in a version FINNv2 to represent regional peat emissions in Indonesia, but this was not included in FINNv2.5]. Similarly, in Equatorial Asia, GFED and GFAS estimate the highest CO emissions for the years when much of the tropical peatland burned. The magnitude of FINNv2.5(MODIS+VIIRS) CO emissions in Australia and New Zealand have increased relative to FINNv1.5, which compares better to downwind surface measurements of instantaneous mixing ratios (Desservettaz et al., 2022). However, regional FINNv2.5(MODIS+VIIRS and MODIS) emissions remain lower than three other emission inventories (GFAS, QFED, and FEER) in Australia and New Zealand. A similar result is seen over the Middle East, which suggests a potential role of extremely dry landscapes in causing inter-inventory differences. For most other regions, FEER is the only inventory that produces emissions that are either as high in magnitude, or sometimes higher, than FINNv2.5(MODIS+VIIRS).

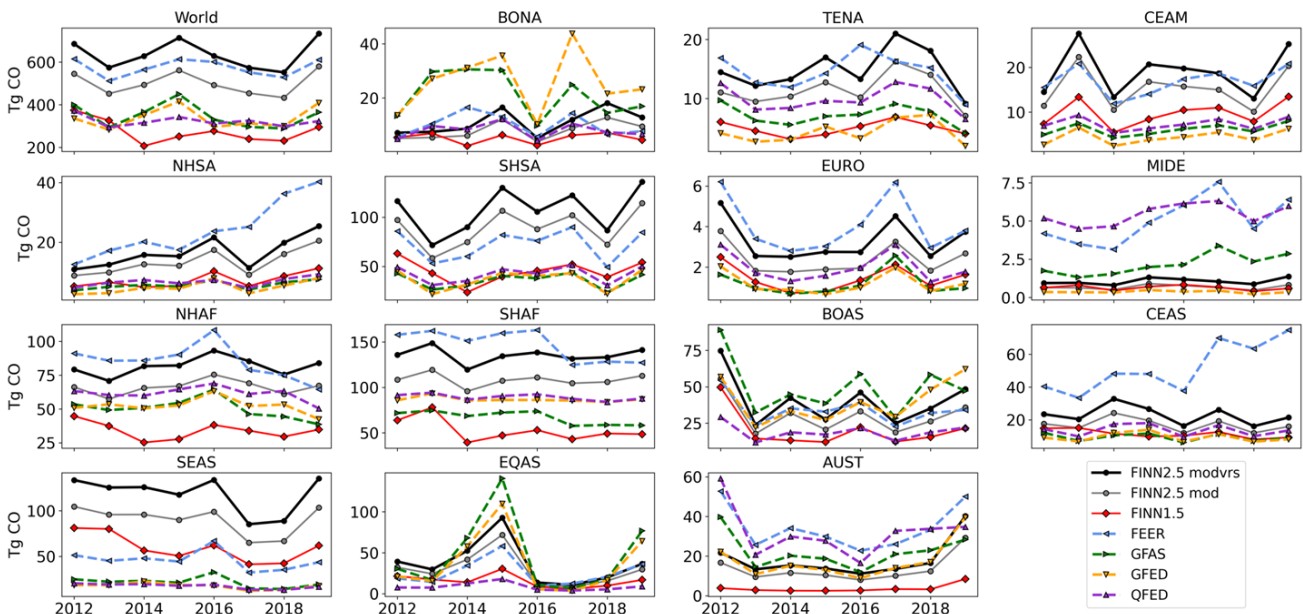

**Figure 5: Annual total emissions of CO by region between 2012 and 2019 from the Fire Inventory from NCAR version 2.5 with MODIS and VIIRS (FINNv2.5 modvrs, black symbols), FINNv2.5 MODIS-only version (FINNv2.5 mod, gray symbols), FINNv1.5, Global Fire Emissions Database (GFED), Fire Energetics and Emissions Research (FEER), Global Fire Assimilation System (GFAS), and QuickFire Emissions Dataset (QFED). Regions are defined in Giglio et al. (2010); refer to Figure 4. For comparisons with earlier versions of FINN, see Figure S3. For comparisons of CH2O and PM2.5 emissions, see Figures S4 and S5, respectively.**

The seasonal change of regional fire emissions for CO in FINNv2.5(MODIS+VIIRS) is shown in Figure 6, with other inventories in Figure S4. Globally, fire emissions peak in August/September, with the largest emissions in Southern Hemisphere Africa and Southern Hemisphere South America. As mentioned above, GFED and GFAS show an increase in Boreal North America in July and August that is not as prevalent in FINNv2.5(MODIS+VIIRS).

FINNv2.5(MODIS+VIIRS) also has an emission peak in March, which is driven primarily by emissions in Southeast Asia. March/April is a peak fire season in the Northern Hemisphere tropics, and in mainland Southeast Asia, the season is driven primarily by small, agricultural fires. FINNv2.5(MODIS+VIIRS) uses VIIRS fire detections, which detect these small fires more so than MODIS. Most inventories show this second peak in emissions during March and April; however, it is not seen in GFED, nor is it as pronounced in the other inventories (Figure S6). Consequently, determining the cause of different fire emissions in Southeast Asia is a target for future research. Although the magnitude of the regional emissions in FINNv2.5(MODIS+VIIRS) is 2-3 times higher than FINNv1.5, the seasonality is similar.

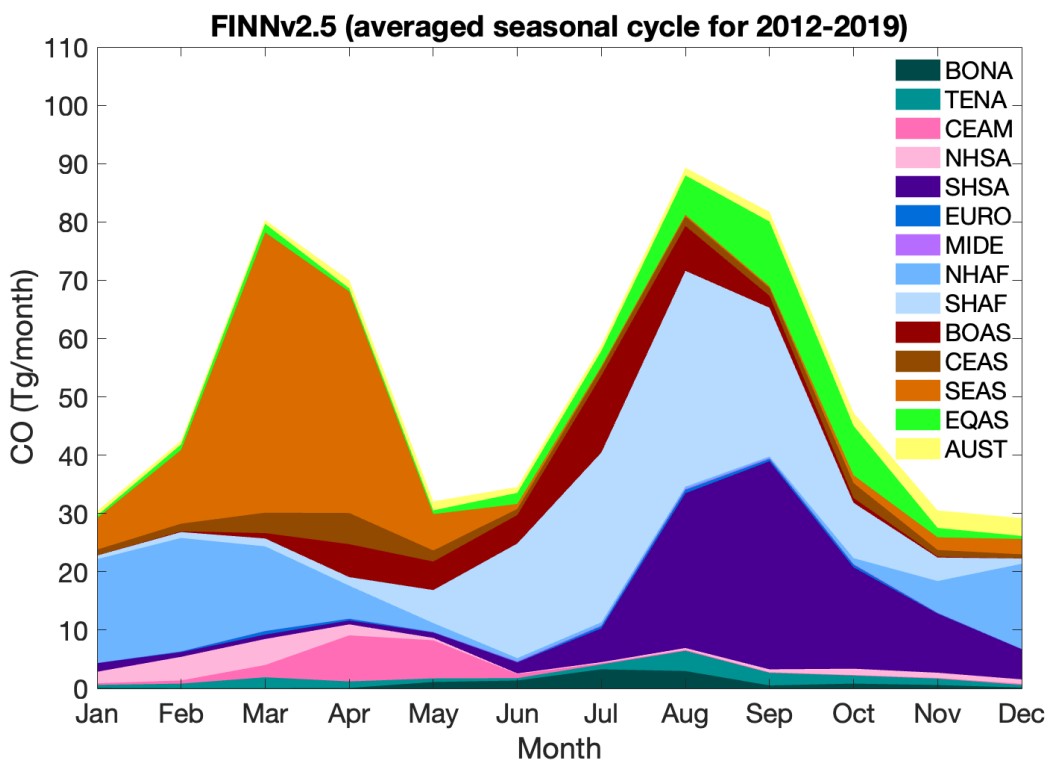

**Figure 6. Seasonal cycle of fire emissions of CO from FINNv2.5(MODIS+VIIRS) (averaged for 2012-2019) by global region. Region definitions follow Giglio et al. (2010) and are described in Figure 4.**

## 3.2 Model simulation using FINNv2.5 compared to satellite products

As shown above, the emission estimates from the different fire emissions models can vary substantially in time and space. It is difficult to know which emission estimates most closely represent reality. One way to assess the emissions is to use them as input to an atmospheric chemistry model and calculate pollutant concentrations that can then be compared to in-situ measurements and satellite observations. To further evaluate FINNv2.5(MODIS+VIIRS) and understand its uncertainties and limitations, we

performed a simulation with the Community Atmosphere Model with Chemistry (CAM-chem), a component of the Community Earth System Model (CESM2.2.0) (Emmons et al., 2020). For this simulation, the temperature and winds of CAM-chem are nudged to the Modern-Era Retrospective analysis for Research and Applications, Version 2 (MERRA-2) reanalysis fields. Anthropogenic emissions are from the Copernicus Atmosphere Monitoring Service (CAMS) v5.1 inventory (Soulie et

al., 2023) and biogenic emissions are calculated online with the Model of Emissions of Gases and

Aerosols from Nature (MEGAN) version 2.1 (described in Emmons et al., 2020). Results for the year 2018, after several years of spin-up, are shown here.

The model results with FINNv2.5 emissions are compared to CO column density from the Measurements Of Pollution In The Troposphere (MOPITT) instrument retrievals (Version 8 Level 3 gridded monthly Joint product; MOP03J.008; Deeter et al., 2019), and Aerosol Optical Depth (AOD) from MODIS (Level 3 gridded monthly global product; MOD08_M3 and MYD08_M3; MODIS Atmosphere Science Team, 2017) (Figures 7 and 8). For the comparisons to the MOPITT retrievals, the model CO profiles are transformed with the MOPITT averaging kernels and an a priori profile. The Joint retrieval product combines the thermal-infrared (TIR) and near-infrared (NIR) radiances to provide greater sensitivity to the boundary layer than the TIR retrievals alone.

Overall, CAM-chem driven with FINNv2.5(MODIS+VIIRS) overestimates satellite-observed AOD and CO over the Amazon Basin and central Africa during the 2018 fire season (Figures 7-9). This result suggests that FINNv2.5(MODIS+VIIRS) overestimates fire emissions over Amazon Basin and central Africa in the 2018 fire season. This overestimation could be due to a number of reasons, including inaccurate ecosystem identification (tropical forest, rather than shrublands or less wooded landscapes) and/or fuel loading assignments. The model also overestimates AOD, but not CO, over Australia and Northern Africa. These discrepancies are likely due to overestimated dust emissions in the model simulation rather than overestimated fire emissions from FINNv2.5(MODIS+VIIRS).

There are important fire regions where the model predictions and observations agree. For example, the CAM-chem results using FINNv2.5(MODIS+VIIRS) simulate column-measured CO in August 2018 (Figure 9) for the Pacific Northwest well. This is consistent with a previous study, which evaluated CAM-chem with regional refinement over the Pacific Northwest with aircraft observations during the Western wildfire Experiment for Cloud chemistry, Aerosol absorption and Nitrogen (WE-CAN) (Tang et al., 2022), and found that simulated CO concentrations agreed reasonably with aircraft measurements.

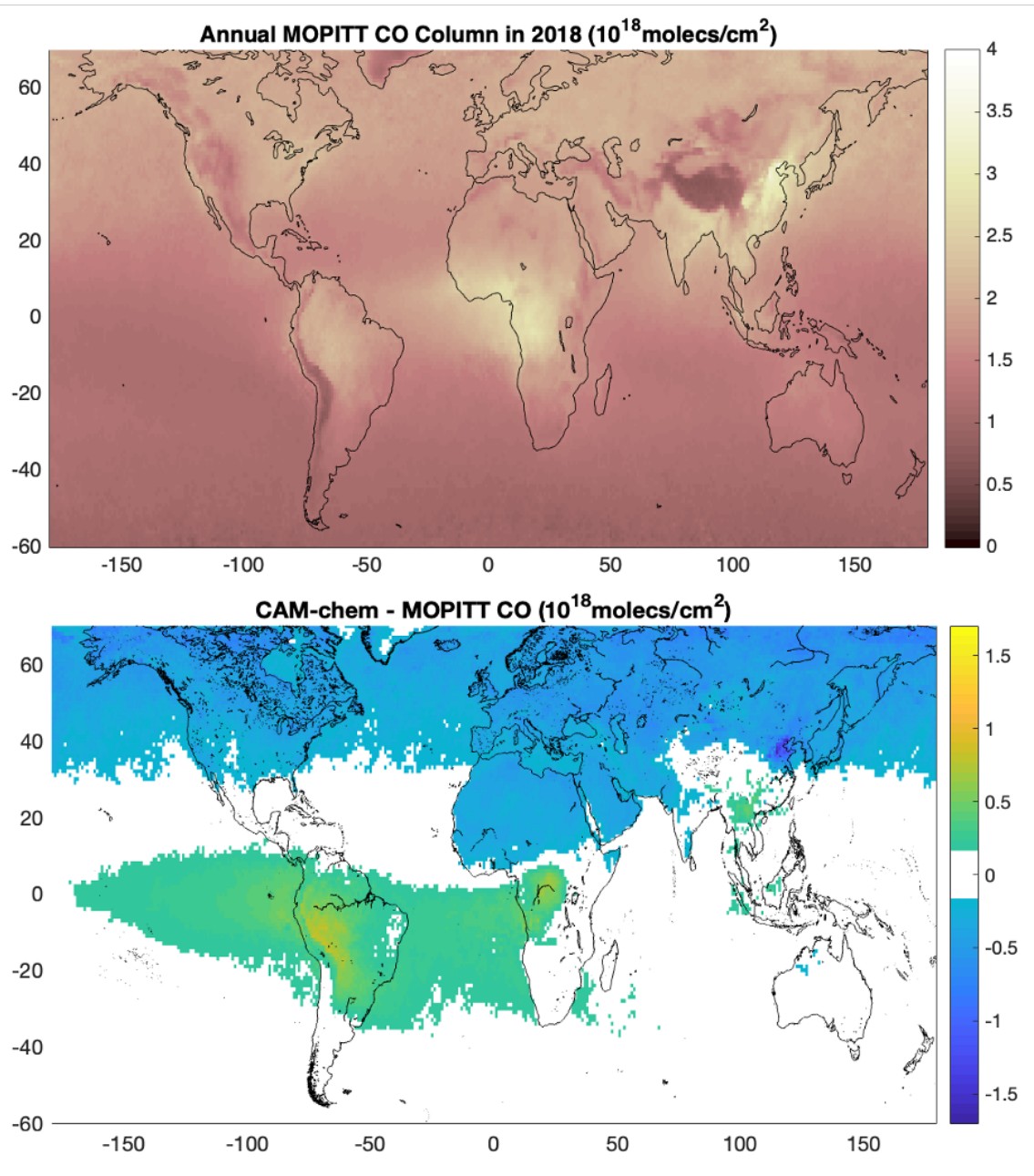

**Figure 7. Global distribution of CO column density from the Measurement of Pollution in the Troposphere (MOPITT) averaged for 2018, and the corresponding Community Atmosphere Model with Chemistry (CAM-chem) model simulation bias (model minus observations) for CO column density. The CAM-chem simulation used FINNv2.5(MODIS+VIIRS) for fire emissions.**

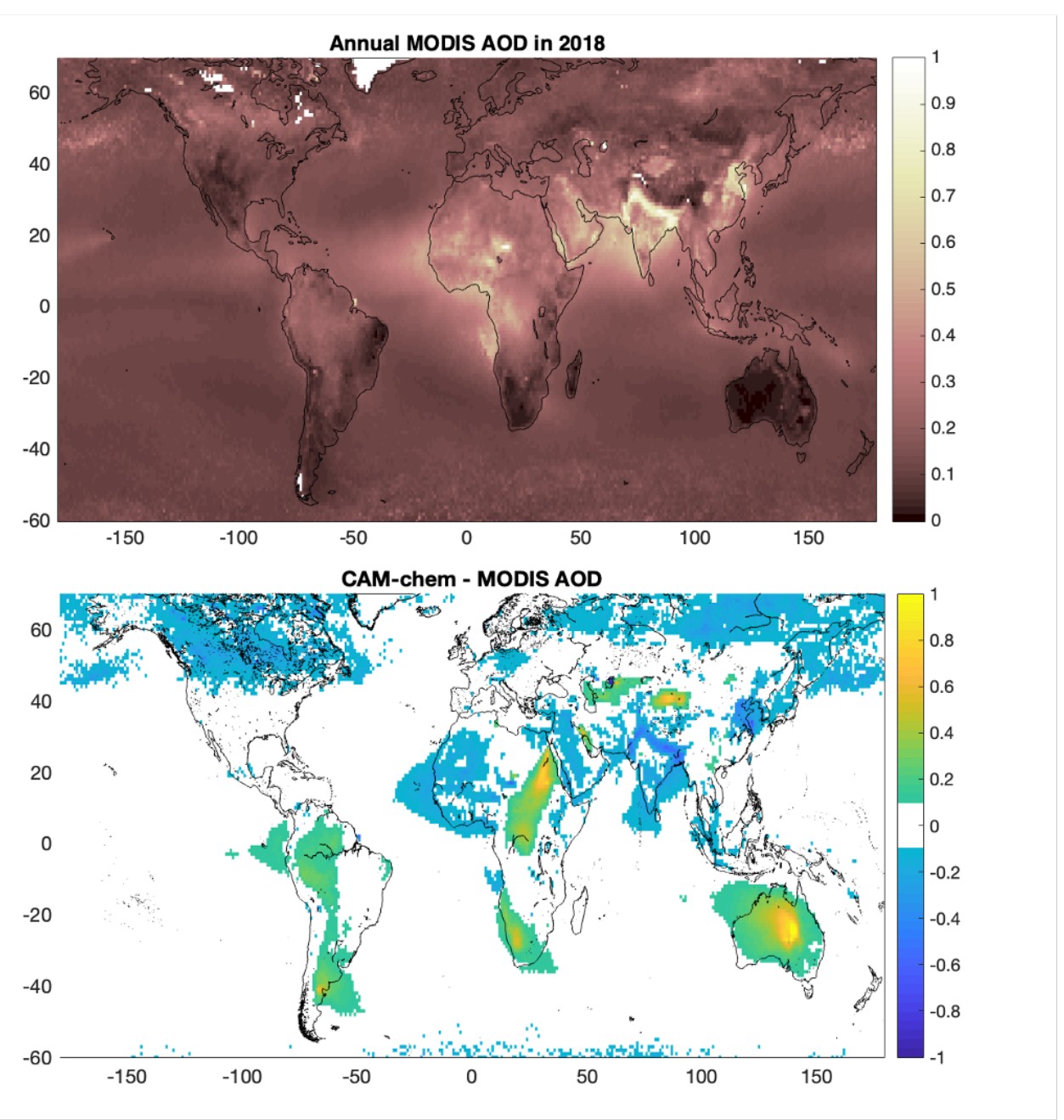

**Figure 8. Global distribution of aerosol optical depth (AOD) from the Moderate Resolution Imaging Spectroradiometer (MODIS) averaged for 2018, and the corresponding Community Atmosphere Model with Chemistry (CAM-chem) model simulation bias (model minus observation) for AOD. The CAM-chem simulation used FINNv2.5(MODIS+VIIRS) for fire emissions.**

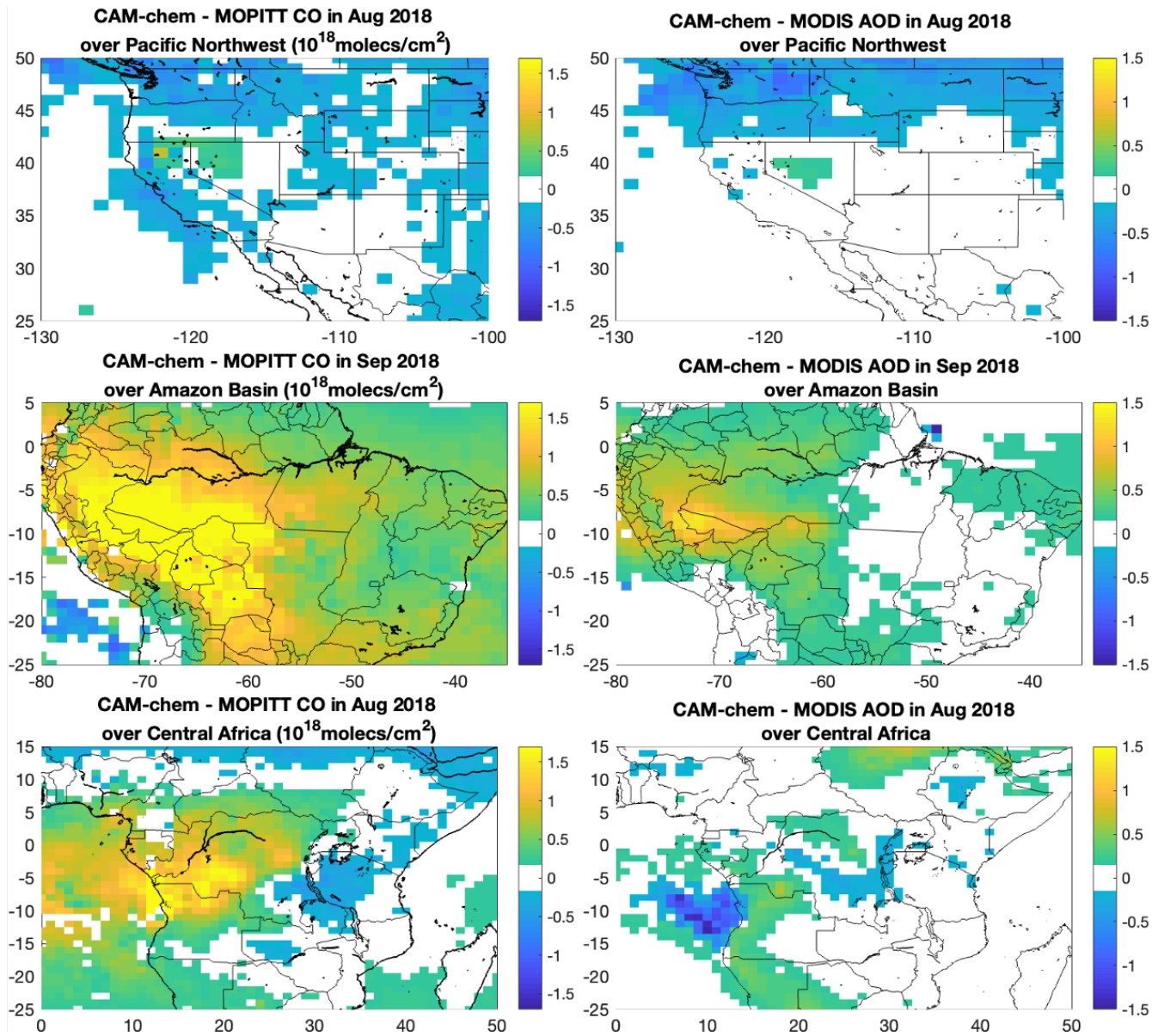

**Figure 9. Community Atmosphere Model with Chemistry (CAM-chem) model simulation bias from MOPITT CO column density (left column) and MODIS AOD (right column) for 3 regions. Top row: the U.S. Pacific Northwest in August 2018; Middle row: Amazon Basin in September 2018; Bottom row: Central Africa in August 2018. The months are selected to represent the fire season in each region. Absolute values modelled with CAM-chem are shown in Figure S7.**

## 3.3 Uncertainties

Despite updates to the input data, parameters, and processing, FINNv2.5 emission estimates remain uncertain. Uncertainties are caused by natural variability associated with the various inputs to the model, and the model assumptions and processes used to create the estimates. Uncertainties may arise due to missed fire detections caused by cloud or smoke cover, timing, and incomplete global coverage from the polar orbiting satellite paths. Further, the assumed vegetation type in active fires are highly variable, and the use of different vegetation maps can introduce large changes in the predicted emissions. The assumed fuel loading, fraction burned, and resulting fuel consumption can be highly variable in space and time, whereas the model assumes best-guess average values for generic ecosystems by global region. The emission factors used also add some uncertainty to the estimates, which is particularly highlighted in the emission estimates of particles and reactive gases. Other issues arise when VIIRS and MODIS are used in combination to drive the emission estimates, compared to the MODIS-only version. We recognize that the addition of a second dataset from VIIRS increases the emissions compared to those estimated using MODIS-only.

Wiedinmyer et al (2011) estimated an uncertainty of a factor of two in the FINNv1 estimates. Other efforts have assigned uncertainty to fire emissions estimates (e.g., Kennedy et al., 2020; Soares et al., 2015; Urbanski et al., 2011); however, limitations in our ability to directly measure fire emissions (fluxes) prevent a comprehensive, global evaluation of existing inventories. Discrepancies between model predictions and evaluations with model output, in-situ measurements, and satellite observations can help identify the processes in the models that drive the uncertainties, and the regions across the globe that are the most important and uncertain. Results from the evaluation presented here suggest that high uncertainties in emissions occur in South America and Southern Africa, and in Southeast and Equatorial Asia. Emissions across boreal North America should also be assessed; wide variations in Organic Carbon (OC) and BC emissions in this region lead to significant uncertainty in the ability to estimate air quality and climate impacts from biomass burning (Carter et al., 2020). Further, while CO and AOD are often

used to assess aerosol and CO emissions estimates, the emissions of other important pollutants are more highly variable across inventories and should be further constrained.

## 4 Conclusions

FINN version 2.5 was created by updating multiple processes and parameters of the original FINN model framework. This version includes improved area burned calculation, using year-specific land cover and
vegetation datasets, updating fuel loading and emission factors, and enabling the use of multiple fire detection satellite inputs (for example MODIS and VIIRS). The python code to process the burned area and overlaid land cover, as well as the IDL code to calculate the emissions and speciate the NMOGs, is freely available to the community for use as is or to be further developed (https://github.com/NCAR/finn-preprocessor, with the current code archived at https://zenodo.org/record/7860860). The resulting
emissions files for 2002-2021 are also freely available in several VOC speciations and gridded formats (Wiedinmyer and Emmons, 2022).

Specific, one-time modifications to FINN have included emissions from peat in Southeast Asia (Kiely et al., 2019) and the consideration of burn severity in the emissions calculations from California (Q. Xu et al., 2022). These may be incorporated into future versions of FINN. Future efforts will also improve
emission estimates for fires in the wildland urban interface.

The FINNv2.5(MODIS+VIIRS) emissions estimates remain uncertain, and more evaluation and model comparisons are recommended, especially in southern hemisphere South America and Africa during August - October, as well as southeast and equatorial Asia in March/April. FINNv2.5(MODIS+VIIRS) does, however, appear to better simulate emissions in the western US, as compared to earlier versions
(e.g., Pfister et al., 2011). The vegetation type and associated fuel loading and consumption are large sources of uncertainty; the use of a different global vegetation map other than the MODIS LCT can lend to large variations in the predicted emissions. Future efforts to improve fire emissions estimates should focus on these components of the model.

**Code Availability**

The code to process the burned area and overlaid land cover, as well as to calculate the emissions and speciate the NMOGs, is available at https://github.com/NCAR/finn-preprocessor, for updated versions, and the current version (v2.5.2) archived at https://zenodo.org/record/7860860. Software to grid the text files for use in 3D models is available on the NCAR/ACOM FINN website (https://www.acom.ucar.edu/Data/fire/).

**Data Availability**

Emissions calculated from the FINNv2.5 algorithms, for MODIS and MODIS+VIIRS fire detections with MOZART VOC speciation, are archived at https://zenodo.org/record/7863959 and https://zenodo.org/record/7868652. The FINNv2.5 emissions, including emissions files for SAPRC and GEOSCHEM VOC speciation, as well as tools for subsetting, are available from the NCAR Research 505 Data Archive (https://rda.ucar.edu/datasets/ds312.9/; https://doi.org/10.5065/XNPA-AF09). Files are provided with emissions for each fire (ungridded, zipped text files) and also gridded to 0.1x0.1 degrees (netcdf files). Both sets of files are available for emissions from MODIS-only and MODIS+VIIRS fire detections.: https://doi.org/10.5065/XNPA-AF09.

**Author Contributions**

CW contributed to all aspects of the project, from model design, management, resource acquisition, model development and evaluation. CW prepared the manuscript with contributions from all co-authors. LKE, WT, RRB assisted in the implementation and evaluation, and led the deployment of the emissions files online. ECMD acquired resources for the development and advised on model development and evaluation. YK let the model development. MJ and KS provided technical support for 515 model development and model simulations. KB, AMC and RY provided key inputs in the model development and evaluation.

**Competing Interests**

The authors declare that they have no conflict of interest.

**Acknowledgements**

The development of the FINNv2.5 model framework, the estimates and the evaluation were supported through several different
funding sources including NOAA CPO Grant NA17OAR4310103; US NSF grant no.1822406; and NASA award
80NSSC18K0681. Funding was also provided by the Texas Air Quality Research Program (AQRP Projects 14-011 and 18-
022) at The University of Texas at Austin through the Texas Emission Reduction Program and the Texas Commission on
Environmental Quality (TCEQ). Support for most CIRES employees is from NOAA award no. NA17OAR4320101 and
NA22OAR4320151. Support for KB was also from NOAA CPO award no. NA17OAR4310007. R. Y. was supported by NSF
Grants AGS-1748266 and AGS-1349976, NOAA-CPO Grant NA16OAR4310100, and NASA Grant NNX14AP45G. The
authors thank the many users of the FINN emission estimates for their support in the evaluation and QAQC efforts.  This
material is based upon work supported by the National Center for Atmospheric Research, which is a major facility sponsored
by the National Science Foundation under Cooperative Agreement No. 1852977. The findings, opinions and conclusions are
the work of the authors and do not necessarily represent findings, opinions, or conclusions of the Texas AQRP or the TCEQ.

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
