# Peer review of "The Fire Inventory from NCAR version 2.5: an updated global fire emissions model for climate and chemistry applications"

_EGUsphere, 2023_

## Author Response (AR1)

**Reply to Editor's comment by Juan Antonio Añel (https://egusphere.copernicus.org/preprints/egusphere-2023-124#CEC2):**

We have now deposited in Zenodo the two primary emissions datasets produced by the FINN2.5 model and presented in the paper, and they are now available at:
https://zenodo.org/record/7863959
https://zenodo.org/record/7868652
The other emissions files mentioned in the paper, for additional VOC speciation and gridded, are derived from these emissions.

The Code and Data Availability sections have been updated in the revised manuscript.
All of the emissions files remain available in the Research Data Archive (RDA) for the convenience of users. The RDA is a domain-specific repository that provides features valued by users, such as subsetting.

The requirement for authentication before downloading has now been removed from the RDA, though this is not listed as a requirement in the Archive Standards of GMD (https://www.geoscientific-model-development.net/policies/code_and_data_policy.html#item3).

We are sorry that the Copernicus journals do not find the RDA to be an acceptable repository for data. We feel it does meet the 3 items listed in the GMD policy, as described in the RDA Terms and Conditions (https://rda.ucar.edu/resources/daas/terms-and-conditions/):

1. The RDA provides a detailed data preservation policy, describing its commitment to providing long-term preservation and access to digital assets. Observations are preserved indefinitely, while model results are preserved for a minimum of 5 years, and are only purged if they have been superseded or not accessed.
2. Contributors to the RDA do not have access to delete their datasets.
3. DOIs are assigned to all datasets, and versions controlled.

We hope in the future that the GMD editors will reconsider which repositories are acceptable for supporting publications.

**Reviewer #1:**

The paper presents an update of the FINN inventory of biomass burning emissions based on satellite-based observations of thermal anomalies. Since FINN provides essential boundary conditions for CTMs and GCMs, the paper addresses a relevant modelling question within the scope of GMD. Furthermore, the calculation of emissions of various chemical smoke constituents from burnt area (observed or modelled) is itself a relevant model parameterisation.

The paper presents a novel concept for calculating burnt area from hot spot observations by the satellite-based MODIS and VIIRS instruments, new compilations of fuel loads and emission factors, the resulting emission data covering 2002-2019 and a comparative assessment thereof. This certainly constitutes sufficient advance to justify a publication.

The methods and assumptions are generally clearly presented. I would just suggest to explain the merging of the data with 2-day resolution into the product with 1-day resolution more clearly, e.g. do the values repeat for pairs of consecutive days? Furthermore, it should be justified that different pixel sizes of the two instruments both can be interpreted as 100% burned; this make sense for clusters of fire pixels but not necessarily for isolated fire pixels.

We are assuming that the reviewer's first point is associated with the lines in the methods stating that "Daily global coverage is not accomplished at latitudes between approximately 23.5° N and 23.5°S due to the observational swath width." This is how the model was set up in earlier versions of FINN (described by Wiedinmyer et al, 2011) and has not changed. We did add a sentence to that section to state "Regardless of whether the detection is from MODIS or VIIRS, the intent is to repeat the fire at the same location since there is not a clear alternative at this time in these global regions, as described by Wiedinmyer et al. 2011." And we do recognize that the 100% burned assumption for all fires is an upper limit to the area, as was assumed in earlier versions. The fraction of fuel in the pixel is scaled based on the type and density of the vegetation. We plan to look further into this assumption in future versions of the model.

The interpretations of the data are somewhat limited due to the lack of reference observations for smoke emissions but sufficiently supported.

The description is complete and precise, in particular since the entire code is published in github. (Personally, I failed to in install the software in macOS due to the error "dpkg-deb: error: 'tini.deb' is not a Debian format archive", though. Help would be appreciated.)

Response: We have moved the code to Zenodo to be compliant with the requirements of GMD. If there are continued issues, we can happily help.

Related work is properly credited and the title reflects the contents of the paper. The abstract is concise and almost complete: I would suggest additionally summarizing the results of the data assessment and the identified regions of largest uncertainties in the abstract and the conclusions.

Response: Thank you for this suggestion. The conclusion does emphasize uncertainties in South America and Africa during August – October, and Southeast/Equatorial Asia in March April. We added text in the abstract to reflect this.

The overall presentation is well structured and clear. Its language is fluent and precise. Symbols and abbreviations are adequately explained except for VCF and LCT, which are, admittedly, names of MODIS products but also are abbreviations.

All parts of the paper should be published and the references and supplementary material are appropriate. Just Fig. 2 doesn't really carry relevant information and may be moved to the online supplement: Fig.2 (c) appears not to be used at all, while (a) and (b) appear to play only minor roles in the following calculations.

Response: Figure 2 has been moved to the supporting information and is now Figure S1.

Finally, here are a few minor suggestions:

L.137: I would state explicitly that this results in a temporal resolution of 1 hour.

Response: Satellite detections of active fires have daily temporal resolution (Section 2.1) except as noted in lines 115-119. The treatment of "local" time is used to assign the day of the detection as well as to facilitate comparisons with ground-based observations. FINN v2.5 provides daily pollutant emissions estimates. We now clarify this at the start of Section 2.

Fig.1(d): The detection rectangles near the bottom are much larger than the 110% mentioned in Line 154. Please clarify the discrepancy in the text.

Response: The "110%" in Line 154 refers to the scan and track size of detection used to identify potentially overlapping detections. In contrast, the blue squares shown in Figure 1(d) (and 1(c)) represent the nominal 1km$^2$ horizontal resolution of the MODIS data (Line 147).  As described in Wolfe et al (1998), the footprint size of the detection is considerably elongated when the MODIS sensor is pointing away from nadir. Hence, the detection coordinates of a fire can be separated.  We assumed that even if the coordinates of two detection points are separated by more than 1 km that they represent a contiguous fire if they are within the scan and track size of the footprint.  The reported distance was not precise enough both in size and orientation of the footprint.  Consequently, we applied an empirical 110% for the scan and track size in order to assure that adjacent sensor detections are identified. The same algorithm was used for both MODIS and VIIRS fire detections.

R. E. Wolfe, D. P. Roy and E. Vermote, "MODIS land data storage, gridding, and compositing methodology: Level 2 grid," in IEEE Transactions on Geoscience and Remote Sensing, vol. 36, no. 4, pp. 1324-1338, July 1998, doi: 10.1109/36.701082.

Fig.1(e): Please replace "VCF" in the legend with the plotted quantity, i.e. tree cover in percent.

Response: The figure has been updated and the legend now reads "Tree cover (%)"

Fig.1(g): The IGBP land cover classes shown here are only introduced in the text in the next section. Is there a smooth way to introduce them early enough for showing in the figure?

Response: We moved Figure 1(g) to Section 2.3 and re-labeled it as Figure 3.

L.221: Since annual MODIS products are used, please state how you are planning to replace them after the lifetime of the MODIS instruments.

Response: This is an excellent point, and one that our team has struggled with. We will be evaluating other land cover and vegetation datasets in the future. FINNv2.5 is an improvement from earlier versions of FINN, in that we use annual specific vegetation, whereas in earlier versions, a static year was used for all simulations. We may have to use the last year of MODIS land cover and VCF for continued years until a new product is available.

L.235: Please summarize the process described in the earlier publication for the convenience of the reader of this paper.

Response: Additional text was added to this section to provide more details.

L.295: Please specify for each of the four instruments which time period it was used for.

Response: All detections from the MODIS and VIIRS instruments for a given day (local time) were used.

Fig.4: I am somewhat surprised that BC emissions are compared but not those of organic carbon/matter. I thought that AOD is usually dominated by the latter and that emission factors for BC are comparatively uncertain. Is there a reason for this choice?

Figure 4 has been updated to show the total BC+OC comparison across inventories. We do think it is valuable to compare BC only, since not all of the inventories have the same BC/OC ratio, and the sum of the two carbonaceous components does make a difference.

Figs.8,9: In the difference plots, it would be nice to distinguish small positive and negative values more clearly, e.g. with small transparent range around zero.

Thank you for this suggestion. We have updated Figures 7, 8, and 9 to more clearly show the differences.

Fig.9: I suggest to also plot absolute value fields, either from the model or the observations.

We have added this as a supporting Figure S7.

I have the impression that "amount that is burned" in L.437 and "fuel consumption" in L.439 refer to the same uncertainty. If so, maybe mention it only once. If not, please clarify what the difference is.

Response: These two are the same. We removed "amount that is burned" from line 437.

L.474: Please justify that FINNv2 "better simulate"s emissions in the western US. This can surely not have been shown in the cited publication from 2011 when FINN2 didn't exist.

Response: We cite Pfister et al. (2011) who highlighted large uncertainties in CO emissions from FINNv1 in California. The new evaluations suggest that FINNv2.5 better represents wildland fire emissions from this region that found in t.

L.480: In github, the version of FINN is specified to be 2.2, with a subdirectory "v2.5_emissions_code". Which parts of the repository/project in github is described by this paper? Please clarify.

Response: Apology for the lag in documentation.   During the review period, we reviewed the repository internally, particularly for documentation.  Latest version as of 2023-04-25, is v.2.5.2, and we will be referring to this version as FINN v2.5.  We have copied the code to Zenodo to comply with the open code and data requirements of GMD. We hope that this removes this issue.

**REVIEWER 2**

The manuscript is generally well written and informative. It presents essential information regarding an important modeling tool for fire emissions and global emissions. I suggest some minor edits to the structure of the presentation as follows:

The Results section (section 3) begins by reviewing what is esentially part of the methods for this manuscript. The comparison to other studies should be presented in the methods section. If desired, the authors could present "model framework" methods section (what is now the Methods) and model comparison methods (move materials to a mehtods section out of Results. Or they could make one Methods section.

Similarly Sec 3.2. presents parts that are actually methods, not resutls (line 384-386: "To further evaluate ..."). I suggest this should also be moved to methods and then the results can be presented as a set of outputs, rather than in the piescemeal way currently presentedd.

Response: We appreciate the reviewer's suggestion here. However, Section 3 (Results) begins with a summary of the outputs of the model and a comparison with other existing model estimates. This is a result from the model described and we believe should remain in this section.

Otherwise, the material presented is excellent and worthwhile.

**REVIEWER Comment 3:**

I have one more practical concern: The label "FINNv2.5" appears to denote the methodology and implementation on the one hand and the resulting emission dataset on the other hand. That is fine. However, two different datasets have been produced from different satellite input. The manuscript refers to datasets as "FINNv2.5" (e.g. Fig. 4 and Line 411), "FINNv2.5(MODIS)" (e.g. Fig. 4) and "FINNv2.5(MODIS+VIIRS)" (e.g. Fig. 7). I recommend to explicitly define which dataset is labelled "FINNv2.5", define another label for the other dataset and use these labels consistently throughout the manuscript.

Response: Thank you for this observation and comment. As you noted, the label FINNv2.5 refers to the overall methodology by which the emissions are calculated. We have edited the manuscript accordingly to clarify the differences.

**REVIEW COMMENT #4**

This paper provides a technical description of the methods used to produce daily global fire emissions estimates from the FINNv2.5 model, which is an update to the widely used FINN model. The most significant updates to the existing model are the inclusion of fire detections from VIIRS, the estimate of event-specific area burned based on geospatial analysis, and updates to fuel characterization and emission factors.

While nothing in the paper is particularly novel, it serves as an important reference to others who will use FINNv2.5 results and should be published, pending some minor clarifications. The paper is well written and clear in most sections. The validation section is reasonable, though it is difficult to assess the uncertainties. The method of modeling CO or AOD to compare with satellite retrievals itself contains many sources of uncertainty, so it cannot be used directly to quantify the accuracy of the fire emissions. This is a shortcoming of the field, and not this specific work. I believe the manuscript could be improved by performing an additional sensitivity analysis using FINNv2.5 with VIIRS detects only.

We agree with the reviewer and the comments here. In light of the end of the MODIS instruments, we recognize the need to move forward with our model development with VIIRS detections only. This is something that we are investing time and effort in, and we will have a revised updated model version in the near future.

Minor comments:

Line 116. How was the value of 20% determined?

Response: This was a value selected in the development of FINNv1 (Wiedinmyer et al., 2011). The value was selected based on inputs from co-authors of the time most familiar with the MODIS Thermal Anomalies product.

Line 124. I would not describe the burning in the southeast US as agricultural burning. It is primarily small-scale prescribed burning on forested land. You could call it "managed burning."

Response: Great suggestion. This edit was made.

Line 145. Can the authors clarify how time is incorporated into this procedure? What time range is used to generate the information in Fig. 1b, for example? Is it a single day? The full year? If it is a single day, is any effort made to link events that overlap in space and are contiguous in time? Are emissions estimated for multiple days if the same location is detected multiple times? If it is a longer time period, how are the emissions allocated back to daily?

Response: FINN v2.5 processes and estimates emissions on a daily basis. Fire detections and fraction of fuel burned for a given day are used in the determination of burned area and associated fuel loading (Sections 2.1-2.3). The approach for all versions of FINN to date does not account for the effects of burning from earlier days, i.e., the fraction burned is consistent with unperturbed vegetation on the first day regardless of the persistence or recurrence of fire activity, and emissions scale directly with the detection of a thermal anomaly in the same location over multiple days. In an example investigation for spring burning in the Yucatan peninsula region (~800 x 1000 $km^2$ area) of Mexico during 2018, we found that approximately 83% of thermal anomalies were detected at most twice in the same location. A cause of at least some second detections could be attributed to "carry over" of fire detections at low latitudes (-23.5 to 23.5 degree) accounted for in the FINN algorithm to compensate for the less than once daily overpass of MODIS in the region. There were, however, cases where fires appeared to occur multiples time in a given vegetated region. For those cases, as described earlier, emissions would scale with detections and could be overestimated. We recommend more comprehensive examination of fire detections globally and the implications of accounting for changes in vegetation due to historical or ongoing fire activity. We added an explanation of our approach in Section 2.1 and note in the caption for Figure 1 that it illustrates the determination of burned area on a specific day

Line 180. This series, especially Fig 1c, suggests that MODIS and VIIRS are seeing different burned areas, or perhaps MODIS is seeing a subset of what VIIRS is seeing, but with less spatial accuracy. A model run using only VIIRS and an analysis of the differences would be very instructive.

Response: For now, we include the MODIS-only fire emissions estimates to provide consistency for our historic emissions estimates (from 2000), and the MODIS-VIIRS dataset. We recognize that the end of the MODIS fire detections will introduce new challenges to the model framework. We believe that combination of the VIIRS and MODIS fire detections provides the most robust inputs to the model, since a single fire is not double counted (each day) and that the inclusion of the two products reduces the missed detections due to cloud cover. Moving forward, we will focus on a VIIRS-only fire emissions dataset and will further evaluate the effectiveness of our burned area calculations.

Line 191. How are the Voronoi polygons determined? What are the points? It appears that they are some subset of the original MODIS/VIIRS detections. What is the purpose of this step? Is it to provide finer spatial resolution?

Response: The subsets of detection points are used as seeds to determine the Voronoi regions, so that the output polygons are approximately similar or slightly greater (when neighboring detection was joined). The reason to choose satellite detection was to facilitate comparison with FINN v1 during development period.  There is no strong reason to choose this algorithm but there is no reason to change either; hence the approach we adapted. The overall objective for applying the Voronoi tessellation algorithm is to provide finer spatial resolution for the final product.

A large fire can span different underlying vegetation types area which results in differences in local emission estimates over the burned area. We recognized this issue and attempted to address it in the development of FINN v2.5. In contrast, emission estimates in FINNv1 were associated with a single detection by MODIS.

Line 305. The differences between FINNv2.5 and FINNv1.5 are significant. The draft mentions that this is "primarily due to new processing of area burned," but can that be further quantified? Clearly emission factors play a large role for some species (e.g., CH2O).

Response: Yes, the differences are significant and the calculation of the area burned and the assignment of the underlying land cover are the biggest drivers. Emission factors also changed, particularly for NMOG, where emission factors in FINNv1 ranged from 5 to 28 g/kg, and in FINNv2.5, they range from 25 to 56 g/kg. The following text was added in the paragraph discussing Figure 4:
"Further, emissions of NMOG and the individual species that make up NMOG (eg., CH2O and C2H6 in Figure 4) are increased significantly due to the use of updated emissions factors from recent field campaigns."

- Line 399. Looks like a typo (...an a priori profile...)

 Response: Corrected.